# A Systematic Analysis of the Correlation between Flavor Active Differential Metabolites and Multiple Bean Ripening Stages of *Coffea arabica* L.

**DOI:** 10.3390/molecules29010180

**Published:** 2023-12-28

**Authors:** Xiaofei Bi, Haohao Yu, Faguang Hu, Xingfei Fu, Yanan Li, Yaqi Li, Yang Yang, Dexin Liu, Guiping Li, Rui Shi, Wenjiang Dong

**Affiliations:** 1Institute of Tropical and Subtropical Cash Crops, Yunnan Academy of Agricultural Sciences, Baoshan 678000, China; rjsbxf@yaas.org.cn (X.B.); y550570@163.com (H.Y.); rjshfg@yaas.org.cn (F.H.); rjsfxf@yaas.org.cn (X.F.); rjslyn@yaas.org.cn (Y.L.); liyaqizzj@163.com (Y.L.); rjsyangy@yaas.org.cn (Y.Y.); susie_liu1995@126.com (D.L.); rjslgp@yaas.org.cn (G.L.); 2Key Laboratory for Forest Resources Conservation and Utilization in the Southwest Mountains of China, Ministry of Education, Southwest Forestry University, Kunming 650224, China; 3Spice and Beverage Research Institute, Chinese Academy of Tropical Agricultural Sciences, Wanning 571533, China

**Keywords:** green coffee beans, UPLC-MS/MS, maturity, flavor, differential metabolites

## Abstract

Coffee cherries contain a crucial flavor-precursor and chemical substances influencing roasted bean quality, yet limited knowledge exists on metabolite changes during cherry ripening. Our study identified 1078 metabolites, revealing 46 core differential metabolites using a KEGG pathway analysis. At the GF vs. ROF stage, amino acid synthesis dominated; ROF vs. BRF featured nucleotide catabolism; BRF vs. PRF exhibited glycoside and flavonoid synthesis; and PRF vs. PBF involved secondary metabolite synthesis and catabolism. The PRF stage emerged as the optimal cherry-harvesting period. A correlation analysis identified core differential metabolites strongly linked to taste indicators, suggesting their potential as taste markers. Notably, nucleotides and derivatives exhibited significant negative correlations with glycosides and flavonoids during ripening. This research systematically analyzed flavor and active substances in green coffee beans during cherry ripening, offering valuable insights into substance formation in *Coffea arabica* L.

## 1. Introduction

The *Coffea* genus, belonging to the Rubiaceae family, is extensively cultivated in various tropical and subtropical regions worldwide. It serves as a significant source of livelihood for millions of individuals residing in coffee-growing nations [1]. *C. arabica* and *C. canephora* are the most important commercial coffee varieties, accounting for about 70% and 30% of the world production, respectively, in agricultural coffee production [2]. The coffee-consuming population in China is growing at an annual rate of 15%, which is more than 7.5 times the global average growth rate. As the largest coffee-producing region in China, Yunnan Province accounts for 99% of the country’s coffee production due to its favorable geographical location and natural conditions. *C. arabica* produced in Yunnan also has a sweet caramel aroma and aromatic acidity [3].

Coffee beverages are popular all over the world for their unique flavor and delightful aroma [4]. Flavor and aroma are the most important criteria in the determination of coffee quality [5] and are also closely related to the biochemical composition of green coffee beans [6]. The biochemical composition of green coffee beans contains many non-volatile compounds. These compounds act as flavor and aroma precursors for many of the volatile compounds produced during the roasting process [7], including amino acids, nucleotides, carbohydrates, lipids, phenolic acids, and terpenoids. The diversity of coffee quality is affected by the biochemical components in green coffee beans accumulated during cherry development [8]. With the ripening of coffee cherries, various components are synthesized from scratch or chemically altered, including the accumulation of solute compounds to store proteins, sucrose, and complex polysaccharides, resulting in distinctive flavors, acidity, and tastes [9]. In past studies, there have been some reports of changes in the chemical composition of coffee cherries during maturation. However, most studies focused on the analysis of markers of immature or overripe beans using different methods. For instance, Amorim performed maturity discrimination of green coffee beans based on electrospray ionization mass spectrometry fingerprinting (ESI-MS) [10]. Craig used diffuse reflectance infrared Fourier transform spectroscopy (DRIFTS) to distinguish immature beans from mature beans [11]. Smrke found that non-volatile compounds were more suitable for differentiating coffee beans of different maturity levels using high-performance liquid chromatography (HPLC) [12]. Hu analyzed the main markers occurring in coffee cherries of different maturity levels using NMR and discussed the effect of maturity marker accumulation on coffee quality in relation to cupping [13]. The above studies focused on changes in the partial biochemical components during the developmental stages of coffee cherries; nevertheless, comprehensive information on the changes in and associations among various metabolites during coffee maturation is more limited.

Metabolomics analysis is a key strategy for analyzing the composition of green coffee beans at the molecular level [14]. The study of metabolites in coffee cherries of different maturity levels is also necessary for understanding the accumulation patterns of biochemical components in green coffee beans. Currently, an ultra-performance liquid chromatography–tandem mass spectrometry (UPLC-MS/MS)-based widely targeted metabolomics analysis has been successfully used for the simultaneous detection of a vast number of metabolites and rapidly provides valuable information on components [15]. Due to the absence of a comprehensive metabolomics analysis of the coffee cherry during maturation, it is difficult to identify and link quantitative changes in nutrients such as amino acids, nucleotides, and carbohydrates in green coffee beans as cherries mature.

Coffee maturity may be described in terms of “days post-bloom” but is more commonly and easily classified by the color. As such, in our study, green coffee beans from five maturity levels were selected based on color to identify and quantify all detected metabolites in green coffee beans with a widely targeted metabolomics analysis using UPLC-MS/MS. With the aid of multivariate statistical and correlation analysis, the synthesis and degradation of nutrients as well as the precursors for flavor and aroma formation in green coffee beans can be clarified, which can be useful for further understanding the accumulation patterns of biochemical components. This study provides valuable information on the nutritional composition of coffee and valuable metabolite data for coffee breeding research.

## 2. Results and Discussion

### 2.1. Widely Targeted Metabolomics Analysis of Coffee Beans at Different Maturity Levels

Before harvesting, a coffee cherry goes through different stages of growth and development, which are the duration of the formation of flavors and active substances. These substances play a vital role in influencing the quality of coffee beverages [16]. Therefore, to better understand the coffee cherry development and ripening process, our study analyzed metabolite changes in green coffee beans at different maturity levels using widely targeted metabolomics based on UPLC-MS/MS. In addition, we focused on the correlation between flavor precursors and taste-related indicators, aiming to provide some reference and insight into the process of coffee flavor substance formation. Changes in color were used to indicate the maturity of coffee cherries (Figure 1A), which were divided into five different maturity stages, GF, ROF, BRF, PRF, and PBF (refer to Figure 1A for a,b,c,d,e, respectively). After the same pre-treatment, the green coffee beans (Figure 1B) were subjected to widely targeted metabolomics analysis. A total of 1078 metabolites were identified, including sugars, amino acids, nucleotides, lipids, and phenolic acids. A large number of these metabolites are flavor and aroma precursors of volatile compounds produced during the roasting process, as well as secondary metabolites with substantial nutritional value and health functions in coffee (Figure 1B and Appendix A). A hierarchical cluster analysis (HCA) showed that all biological replicates from the same maturity were clustered together, denoting the reliability of the metabolic profiling data (Figure 1C). Among these, the samples from the five different metabolites were clearly separated, which implies that a large number of metabolites are differentially accumulated between these samples.

### 2.2. Multivariate Analysis of the Identified Metabolites

To examine all the metabolomic differences among the five different maturity coffee beans and the variability in the GF, ROF, BRF, PRF, and PBF groups, the data set from the 1078 metabolites was analyzed using principal component analysis (PCA). All samples were mixed in equal volumes with three replicates for quality control (QC) purposes. As shown in Figure 2A, three QCs clustered in the center, indicating a good run result. Also, the contributions of PC1 and PC2 were 30.94% and 15.06%, respectively. No significant separation occurred between the GF and ROF groups, showing relatively slow cherry development in the unripe to early stages of color change. The obvious separation of the BRF, PRF, and PBF groups indicated that the cherries entered the rapid development stage. It also can be observed that the PRF group was distinguished from the BRF and PBF group by PC2. The results of our study showed good reprehensibility across replicates in the same group of samples and different metabolite profiles for samples of different maturity.

To further determine the differences in metabolite composition between the samples, we amplified metabolite differences between groups using a partial least squares-discriminant analysis (OPLS-DA) model to assess the cherry samples at different levels of maturity. This included GF versus ROF (R2X = 0.508, R2Y = 0.999, Q2Y = 0.937), ROF versus BRF (R2X = 0.719, R2Y = 0.996, Q2Y = 0.945), BRF versus PRF (R2X = 0.751, R2Y = 1, Q2Y = 0.992), and PRF versus PBF (R2X = 0.732, R2Y = 0.999, Q2Y = 0.939). The Q2Y values among all subgroups with different maturity exceeded 0.9, which indicates that the OPLS-DA model analysis was stable and reliable. These results also exhibit that the VIP values obtained could be used to screen for differential metabolites.

### 2.3. Identification and Classification of Differential Metabolites

To ascertain differential metabolites (DMs) in green coffee beans occurring significantly with cherry ripeness, we identified a total of 260 DMs in four comparison groups based on the screening conditions of |log_2_FC| ≥ 1 or |log_2_FC| ≤ −1 and VIP ≥ 1. Among these, there were 56, 80, 167, and 78 DMs in GF versus ROF, ROF versus BRF, BRF versus PRF, and PRF versus PBF, respectively (Figure 3B). These DMs mainly include amino acids and derivatives, organic acids, alkaloids, terpenoids, lipids, etc. As shown in Figure 3A, the least number of DMs in the GF to ROF stage indicated slow changes in cherry composition, which is consistent with the results of the PCA analysis (Figure 2A). We also found the only DM—Procyanidun B2—in the GF to ROF stage. The majority of the proanthocyanidins are synthesized by the flavonoid metabolic pathway, causing the cherry to exhibit a pronounced sour taste [17]. It was shown that the sourness and astringency of coffee decreased significantly during the later stages of cherry development. Nucleotide and derivative DMs were significantly higher in the ROF to BRF stage. In addition, the BRF to PRF stage was the period of fastest improvement in cherry quality, with a markable increase in organic acid, lipid, and saccharide DMs that contribute to sweetness, acidity, and softness. Furthermore, we detected several vital physiologically active substances in green coffee beans, such as caffeine, various chlorogenic acids (3-*O*-caffeoylquinic acid, 4-*O*-caffeoylquinic acid, 5-*O*-caffeoylquinic acid, etc.) and trigonelline. These vital active substances remain relatively stable throughout the growth and development stages of coffee cherries. Chlorogenic acid and caffeine are usually responsible for the bitter taste of coffee. For human health, caffeine has several neuroprotective properties [18], while chlorogenic acid has important effects in terms of antioxidant activity, anti-diabetic, hepatoprotective, hypoglycaemic, and antiviral properties [19,20,21]. Both of them form a 1:1 complex through hydrophobic interactions [22]. Trigonelline is a pyridine derivative with an average content of about 1% in *Coffea arabica* L., and it has neuroprotective and anti-inflammatory effects on human health [23].

### 2.4. Kyoto Encyclopedia of Genes and Genomes (KEGG) Enrichment Analysis of Differential Metabolites

To further clarify the biological pathways involved in the significant enrichment of DMs during cherry ripening, we performed a KEGG enrichment analysis. According to the KEGG annotations, 20 pathways were significantly enriched in total for the five stages (*p* < 0.05). As shown in Figure 4A, for the GF stage versus the ROF stage, the representational enriched terms were “cysteine and methionine metabolism”, “zeatin biosynthesis”, “sulfur metabolism”, “plant hormone signal transduction”, “caffeine metabolism”, etc. The synthesis and metabolism of amino acids and derivatives are very active at this stage. Amino acids can be used as one of the flavor precursors to directly produce sweetness, freshness, and so on. In our research, amino acids and precursors such as O-acetyl serine, N-formyl-L-methionine, 5′-deoxy-5′-methylthioadenosine, and L-cystine were significantly increased (Figure 5A). That indicates active synthesis of cysteine, cystine, and methionine. Most of them are volatile and strong olfactory substances. Among them, methionine produces maltiness and fruitiness, while cysteine and cystine are important components of cooked meat and butter aromas, which strongly influence the flavor of coffee beverages [24].

For the ROF stage versus the BRF stage, the representational enriched pathways were “nucleotide metabolism”, “purine metabolism”, “zeatin biosynthesis”, “pyrimidine metabolism”, “ABC transporters”, etc. Nucleotides and derivatives were actively metabolized at this stage (Figure 4B), with significant decreases in the content of nucleotides and precursors such as xanthosine, guanosine, cytidine, guanine, guanosine 5′-monophosphate, 2′-deoxyguanosine, 2′-deoxyadenosine, and succinyladenosine, and other nucleotides. The precursors were significantly reduced in content as well. Of these, xanthosine, guanosine, and cytidine are all typical flavor-presenting nucleotides, combining with monosodium glutamate to exhibit typical fresh flavors [25]. Moreover, they have a potentiating effect on sweetness, meatiness, and mellowness and inhibit flavors such as sourness, bitterness, and burntness [26]. During the ripening of coffee cherries, the decrease in the content of the presenting nucleotides significantly enhances the development of flavor precursors such as acidity, bitterness, and tartness in the coffee.

For the BRF stage versus the PRF stage, the representational enriched pathways were “pentose and glucuronate interconversions”, “amino sugar and nucleotide sugar metabolism”, “flavone and flavonol biosynthesis”, “biosynthesis of nucleotide sugars”, “galactose metabolism”, etc. We detected that various saccharide synthesis, metabolism, and flavonoid synthesis pathways were dramatically enriched. As shown in Figure 4C, at this stage, saccharides such as D-glucose, D-fructose, D-mannose, D-arabinose, D-galactose, and L-xylose were increasing significantly. These saccharides affect the volatile flavor components produced by the coffee during the brewing process and help to enhance the floral and caramel flavors of the coffee beans after roasting [27]. A study conducted ^1^HNMR metabolomics on eight coffee beans of different maturity levels and found that saccharides increase dramatically in the flesh during late cherry development [13]. In addition, saccharides could be used as markers of coffee cherry maturity, but not clearly in green coffee beans. Our research showed that the saccharide content of green coffee beans increases significantly in the later stages of cherry development, and the coffee ripens as the color of the cherry turns completely red. Meanwhile, flavonoids such as lonicerin and vicenin-2 and flavonols such as rutin, narcissin, quercetin-3-*O*-rutinoside-7-*O*-glucoside, and limocitrin-7-*O*-glucoside were significantly increased at this stage. These substances are present in the green coffee bean as bound glycoside compounds, which are volatilized at high temperatures during the roasting process to enrich the aroma of coffee [28]. Furthermore, a high intake of flavonoid substances can be of great benefit to human health. For instance, flavonoids such as lignans and quercetin have strong antiviral and bactericidal as well as free radical scavenging and antioxidant effects [29]. For this reason, the PRF stage can be considered the best harvesting period for coffee cherries.

For the PRF stage versus the PBF stage, the representational enriched pathways were “monoterpenoid biosynthesis”, ”caffeine metabolism”, “plant hormone signal transduction”, “zeatin biosynthesis”, ”biosynthesis of secondary metabolites”, etc. From the above, we obtained 46 differential metabolites in significantly enriched KEGG pathways. As shown in Figure 4D, at this stage, the terpenoids and caffeine metabolic pathways were significantly enriched for a large number of secondary metabolites such as eniposidic acid, uncargenin D, shanzhiside, etc. A large number of terpenoids were present and could add aroma to coffee. At present, the studies on the terpenoids of coffee mainly focus on caffeine and cafestol. The presence of 16-*O*-methylcafestol exclusively in *C. arabica* makes it a reliable marker for distinguishing *C. arabica* from *C. canephora* [30]. Moreover, these terpenoids have been shown to have positive effects in the study of a variety of diseases related to human health [31,32]. Apart from terpenoids, we also found significant changes in the theophylline and theobromine in the caffeine metabolic pathway during this stage. The increase in theophylline content was accompanied by a decrease in theobromine content. In contrast, theophylline has a stronger central euphoric effect and effectively improves symptoms such as asthma and bronchitis.

### 2.5. Correlation Analysis of Characteristic Metabolites during Cherry Development and Ripening

The KEGG pathway analysis clarified 46 DMs changes during coffee cherry development and ripening. The metabolites from the five developmental stages were grouped into five groups using a hierarchical cluster analysis (Figure 5A). The metabolites in groups 1 and 2 were reduced from GF to PBF, including 14 nucleotides and their derivatives, four organic acids, two monoterpenes, vitamin B2, and theobromine. The metabolites in groups 4 and 5 were increased from BRF to PBF. Among these, metabolites in group 4 include 2-hydroxyethylphosphonic acid, cynaroside, rutin, D-glucoronic acid, and D-galacturonic acid, while metabolites in group 5 have all amino acids and their derivatives, lactose, 3-methylxanthine, and theophylline. In contrast, the metabolites in group 3 showed a typical increase during cherry development and ripening. Group 3 contained 11 saccharides (glucose-1-phosphate, D-glucose-6-phosphate, D-glucose, D-arabinose, D-mannose, D-galactose, xylitol, D-sorbitol, D-mannitol, D-arabitol, and dulcitol) and two flavonoids (nicotiflorin and lonicerin) that increased rapidly during BRF to PRF.

To eliminate the effects of quantity on pattern recognition, we applied a log_2_ transformation of peak areas for 46 DMs, followed by a correlational analysis between them and the taste indicators [33]. As shown in Figure 5B, the target 46 DMs correlated strongly with 11 of the 27 taste indicators (*p* < 0.05, Pearson’s r > 0.8), including one sugar, nine amino acids, one chlorogenic acid, and trigonelline (Appendix A). Fructose is important in green coffee. It had a positive correlation with saccharide metabolites and flavonoid metabolites. Cryptophyllogenic acid had a positive correlation with lactose. Trigonelline had a negative correlation with dulcitol and D-sorbitol, and GABA had a positive correlation with dulcitol and xylitol. Glu had a negative correlation with various nucleotide metabolites, while all other amino acids had a positive correlation with various nucleotide metabolites as well as organic acid metabolites, and a negative correlation with sugar metabolites and flavonoid metabolites. Among these, Trp acts as a discriminator between mature and immature green coffee beans [34] and plays a crucial role during coffee cherry development. In our research, the highest Trp content was found during the GF period and gradually decreased with coffee cherry maturity, consistent with the results obtained in this research. The correlation analysis (Figure 5B) revealed that Trp was significantly negatively correlated with theophylline, 3-methylxanthine, and D-mannitol and positively correlated with 7-deoxyloganic acid, adenylocuccinic acid, abscisic acid, and various nucleotides and derivatives. We speculate that these substances could be used as candidate markers of coffee ripening, pending further studies.

The correlations between the target 46 DMs are shown in Figure 5C, with a strong positive correlation between saccharides and flavonoids, as well as a strong negative correlation with nucleotides and derivatives. While nucleotides and their derivatives had a significant positive correlation with organic acids and vitamins and a significant negative correlation with alkaloids. However, the impact of ripening-induced changes in these compounds on baked beans remains unknown and requires further investigation. Additionally, it is essential to enhance our understanding of diversity by considering different coffee varieties with distinct flavor characteristics and varying levels of flavor precursors in green coffee beans.

## 3. Conclusions

In our research, a thorough metabolite detection analysis of green coffee beans of different maturity levels was carried out using widely targeted metabolomics based on UPLC-MS/MS. The analysis identified 260 DMs with 56 (GF versus ROF), 80 (ROF versus BRF), 167 (BRF versus PRF), and 78 (PRF versus PBF) in four different stages, respectively. Then, the synthesis and metabolism of the main flavor active substances in green coffee beans at different stages of maturity were clarified using a DM analysis and KEGG pathway analysis. For the GF stage versus the ROF stage, the synthesis of amino acids and derivatives were mainly identified; for the ROF stage versus the BRF stage, the catabolism of nucleotides and derivatives were mainly identified; for the BRF stage versus the PRF stage, the synthesis of glycosides and flavonoids were identified; for the PRF stage versus the PBF stage, the synthesis and catabolism of secondary metabolites were identified. In addition, our research selected several potential cherry ripening markers with a correlation analysis of target DMs with typical coffee flavor indicators, including 7-deoxyloganic acid, adenylocuccinic acid, abscisic acid, and various nucleotides and derivatives. Further validation of these potential coffee ripening marker compounds is warranted in future studies. Furthermore, our correlation analysis between metabolites clarified the correlation among different components of the DMs during coffee cherry ripening. Overall, the results of this research systematically analyzed the flavor and active substances in green coffee beans during cherry ripening, providing valuable reference information to clarify the formation of substances in *Coffea arabica* L.

## 4. Materials and Methods

### 4.1. Plant Materials and Test Condition

The test material used in this study was the *C. arabica* cultivar *Catimor.* Quality seeds were planted at the scientific research base of the Institute of Tropical and Subtropical Cash Crops of Yunnan Academy of Agricultural Sciences in Lujiang Town, Baoshan City (24°58′17.93″ N; 98°52′43.28″ E), Yunnan, China. This area has an altitude of 750 m and belongs to the low-latitude quasi-tropical monsoon forest. It has a dry and hot valley transition type of climate with sufficient sunlight and a small annual temperature difference. There is no frost throughout the year. The annual mean temperature, rainfall, and air humidity are 21.5 °C, 755.3 mm, and 70%, respectively. Five cherry-colored coffee plants were selected from the same cultivar, and at least twenty plants for each cherry color were planted in four rows by maintaining a 1 m plant-to-plant and 2 m row-to-row distance. Five random plants were selected to perform sampling. Finally, five fully mature and even-colored fresh cherries were harvested for each color under study (Figure 1A). Each harvested cherry was placed in a specialized net, labeled, and subjected to pre-treatment. That is, the fresh cherry was peeled with a peeling machine. Following the peeling process, the freshly harvested cherry was immersed in clean water for a duration of 48 h. Once the pectin was thoroughly cleaned and there were no traces of stickiness or slipperiness, it was removed from the water and allowed to dry naturally. The moisture content of the green beans was swiftly determined using a moisture measuring instrument. When the moisture of the green beans was 10–12%, metabolite determination was carried out quickly.

### 4.2. Sample Preparation and Extraction

The samples were freeze-dried using a freeze-dryer (Scientz-100F, Xinzhi Freeze Drying Equipment Co., Ltd., Ningbo, China) and then ground (30 Hz, 1.5 min) into a powder using a grinder (MM 400, Retsch, Dusseldorf, Germany). Then, 50 mg of sample powder was weighed using electronic scales (MS105DM, Mettler Toledo, Shanghai, China) before adding 1200 μL of 70% methanol aqueous internal standard extract (−20 °C, 2-chlorophenylalanine, 1 PPM) for extraction and vortexing. Then, centrifugation (5424R, Eppendorf, Hamburg, German) was performed at 12,000 rpm for 3 min. The supernatant was aspirated, filtered through a microporous membrane (0.22 μm pore size), and finally stored in an injection vial until UPLC-MS/MS analysis.

### 4.3. UPLC Conditions

The sample tuber extracts were analyzed using a UPLC-ESI-MS/MS system (UPLC, ExionLC™ AD, https://sciex.com.cn/, accessed on 23 March 2022; MS/MS, applied Biosystems 4500 QTRAP, https://sciex.com.cn/, accessed on 1 April 2022). The analytical equipment and solutions were as follows: UPLC: the column was Agilent SB-C18 (1.8 µm, 2.1 mm ∗ 100 mm); the mobile phase consisted of ultra-pure water (0.1% formic acid) for solvent A and acetonitrile (0.1% formic acid) for solvent B. The elution gradient started at 0.00 min with a B phase proportion of 5%. Over the course of 9.00 min, the proportion of the B phase increased linearly to reach 95% and remained constant for an additional minute. From 10.00 to 11.10 min, the proportion of the B phase decreased back to 5% and maintained this level until reaching 14 min. The flow rate, column temperature, and injection volume were set to 0.35 mL/min, 40 °C, and 4 μL, respectively. The effluent was alternatively connected to an electrospray ionization (ESI)-triple quadrupole-linear ion trap (QTRAP)-MS.

### 4.4. ESI Q-TRAP-MS/MS

The ESI source operation parameters were as follows: ion source, turbo spray; source temperature, 550%; ion spray (IS) voltage, 5500 V (positive ion mode)/−4500 V (negative ion mode); ion source gas I (GSI), gas II (GSII), and curtain gas (CUR) were set to 50, 60, and 25.0 psi, respectively; the collision gas (CAD) was high. Instrument tuning and mass calibration were performed with 10 and 100 μmol/L polypropylene glycol solutions in the QQQ and LIT modes, respectively. QQQ scans were acquired as multiple reaction monitoring (MRM) experiments with the collision gas (nitrogen) set to medium. To produce maximal signals, the collision energy (CE) and declustering potential (DP) were optimized for individual Multiple Reaction Monitoring (MRM) transitions. A specific set of MRM transitions was monitored for each period according to the metabolites eluted within the period.

### 4.5. Qualitative and Quantitative Analysis of Metabolites

Based on the Metware metabolic database (Metware Biotechnology Co., Ltd., Wuhan, China), the metabolites in the samples were qualitatively and quantitatively analyzed using mass spectrometry. The characteristic ions of each substance were screened with the triple quadrupole, and the signal intensity of the characteristic ions was obtained in the detector by count per second (CPS). The downstream mass spectra of the samples were opened using MultiQuant software (V3.0.3) to carry out the integration and correction of the chromatographic peaks, where the area of each peak represented the relative content of the corresponding substance. Finally, all chromatographic peak area integral data were derived. To compare the substance content of each metabolite in different samples for all the metabolites detected, the peaks of each metabolite detected in different samples were calibrated for qualitative and quantitative accuracy based on information on metabolite retention times and peak shapes.

### 4.6. Metabolites Analysis

The hierarchical cluster analysis (HCA) of the sample metabolites is a categorical multivariate statistical analysis, with the highest possible heterogeneity between categories. Principal component analysis (PCA) is an unsupervised pattern recognition method of statistical analysis of multidimensional data, which was used to provide a preliminary understanding of metabolite differences between groups of samples and the magnitude of variability between samples within groups. Orthogonal partial least squares discriminant analysis (OPLS-DA) was used to maximize metabolic differences between the two samples using a supervised multivariate approach, using variable importance in projection (VIP) to examine the relative importance of each metabolite in the OPLS-DA model. Among these, metabolites with |log_2_FC| ≥ 1 or |log_2_FC| ≤ −1 and VIP ≥ 1 were considered as differential metabolites (DMs) between groups. Ward’s hierarchical clustering heatmap (HCA), principal component analysis (PCA), orthogonal partial least squares discriminant analysis (OPLS-DA), and Pearson’s correlation analysis were performed in the R software version 4.2.3 (www.r-project.org, accessed on 7 January 2023).

### 4.7. KEGG Annotation and Enrichment Analysis

The identified metabolites were annotated using the KEGG database (http://www.kegg.jp, accessed on 14 January 2023) and then mapped to KEGG pathways. Differential metabolite-enriched pathways were fed into metabolite set enrichment analysis (MSEA), and their significance was determined with the hypergeometric test’s *p*-values.

### 4.8. Quantification of Taste Indicators

Selected typical taste-related indicators of importance in coffee, including amino acids (GABA, Glu, Phe, Tyr, Arg, Val, Met, Asp, Asn, His, Ile, Leu, Lys, Pro, Thr, and Trp), chlorogenic acid (neochlorogenic acid, cryptochlorogenic acid, chlorogenic acid, isochlorogenic acid A, isochlorogenic acid B, isochlorogenic acid C), alkaloids (caffeine, cafestol, trigonelline), and saccharides (sucrose, fructose), were analyzed with UPLC-MS/MS (Appendix A). All analyses were repeated three times.

### 4.9. Statistical Analysis

Data quality and reproducibility of the metabolites detected were assessed using principal component analysis (PCA), hierarchical clustering analysis (HCA), and Pearson’s correlation analysis based on ion intensities with the tidyverse, pheatmap, and corrplot packages in R (v3.3.2), respectively. The bar plot was created using GraphPad Prism 8 (GraphPad Software Inc., San Diego, CA, USA). The differential metabolites (DMs) were identified using stringent filtering criteria of the orthogonal partial least squares-discriminant analysis (OPLS-DA) using a threshold of log_2_ foldchange, |log_2_FC ≥ 1|, and variable importance in projection (VIP) ≥ 1. The heatmap of selected DMs was constructed in R (v3.3.2) with the pheatmap package.

## Figures and Tables

**Figure 1 molecules-29-00180-f001:**
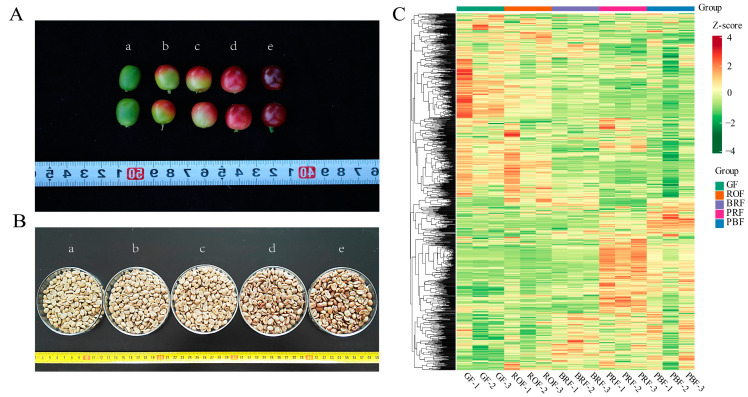
(**A**) Phenotype characteristics of coffee cherry at different ripeness. (a) The unripe stage with a green color. (b) The color breaker stage with a reddish-orange color. (c) Successive ripening and pigmentation deepening with a bright red color. (d) The ripe stage with a purple-red color. (e) The overripe stage with a purple-black color. (**B**) Green coffee beans at different ripeness. Lowercase letters (a–e) correspond to (**A**) for green coffee beans in the fresh cherry of different ripeness. (**C**) Hierarchical cluster analysis (HCA) of metabolites in coffee bean samples with different maturities, from low (green) to high (red). The Z-score represents the deviation from the mean by standard deviation units.

**Figure 2 molecules-29-00180-f002:**
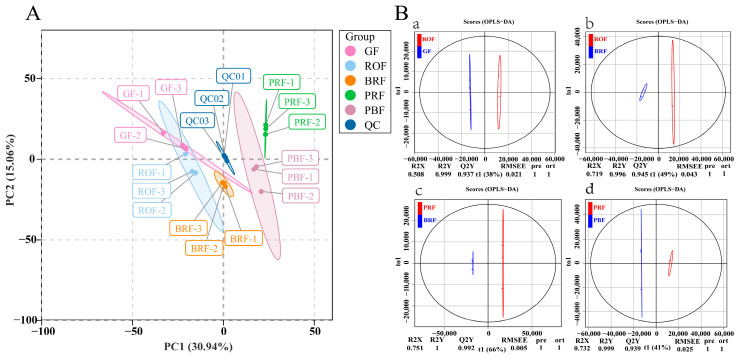
(**A**) Principal component analysis (PCA) of metabolites in coffee beans at different maturity levels. Note: Each maturity level had three individual samples. Equal volumes of samples were mixed for use as the quality control (QC). (**B**) OPLS-DA of metabolites in five different maturity treatments among GF, ROF, BRF, PRF, and PBF. (a) GF versus ROF. (b) ROF versus BRF. (c) BRF versus PRF. (d) PRF versus PBF.

**Figure 3 molecules-29-00180-f003:**
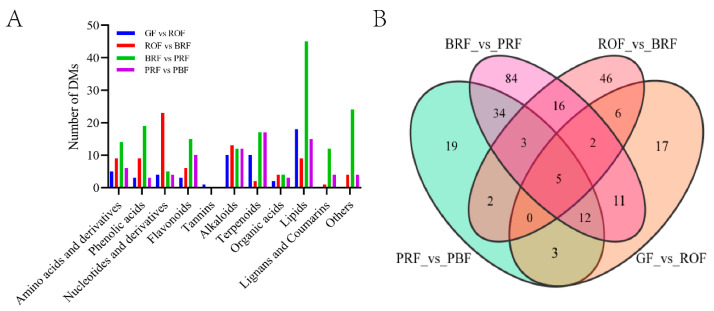
(**A**) Classification and statistics of DMs in green coffee beans at five ripeness. Others mainly include saccharides and vitamins. (**B**) A Venn diagram of DMs.

**Figure 4 molecules-29-00180-f004:**
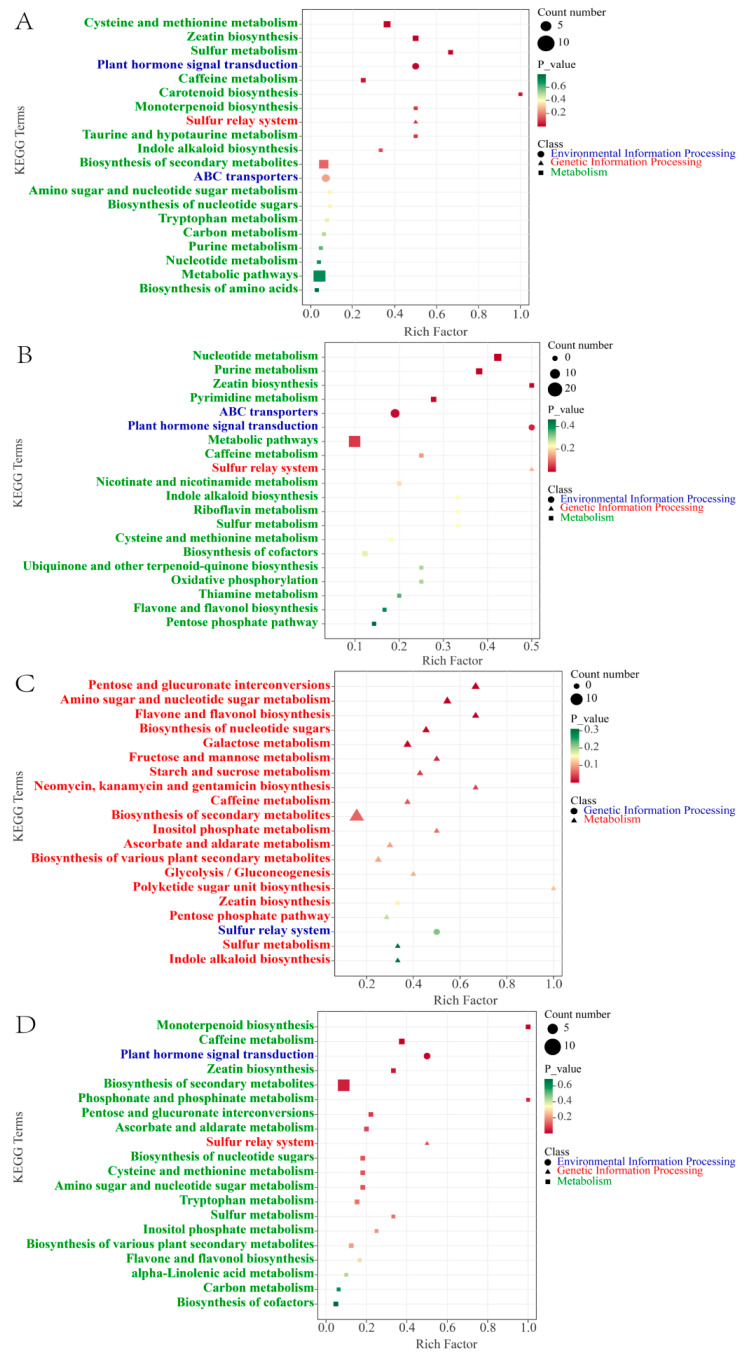
KEGG enrichment analysis of DMs. Top 20 KEGG pathways of GF versus ROF (**A**), ROF versus BRF (**B**), BRF versus PRF (**C**), and PRF versus PBF (**D**).

**Figure 5 molecules-29-00180-f005:**
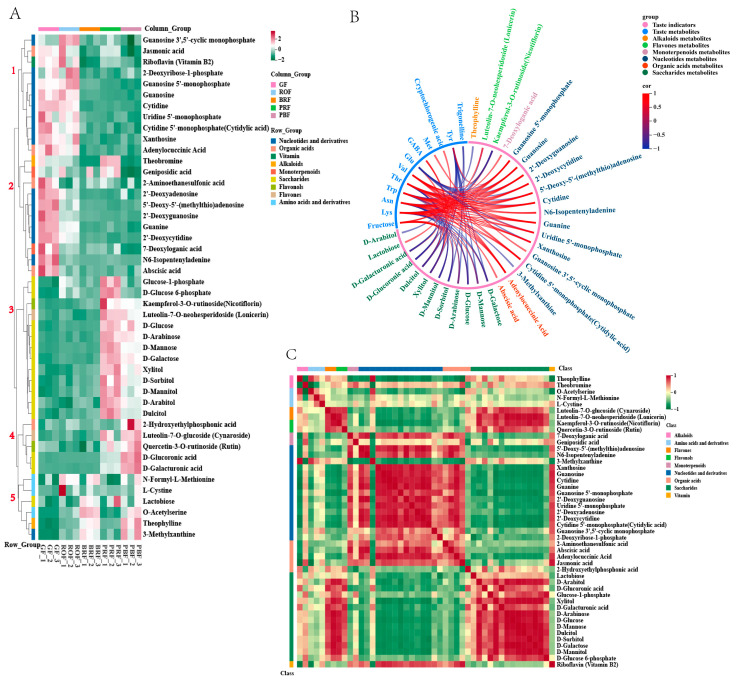
Clustering analysis of 46 target metabolites in green coffee beans at five maturity levels; the metabolites are grouped by compound class. (**A**) Hierarchical clustering analysis of metabolites and different maturity levels; five main branches were generated. (**B**) Correlation between taste indicators and metabolites; the correlation coefficients were calculated using Pearson’s test (Pearson’ r > 0.8, *p* < 0.05). (**C**) Visualizations of metabolite–metabolite correlations were calculated using a Pearson correlation analysis (*p* < 0.05).

## Data Availability

The data presented in this study are available upon request from the corresponding author. The data are not publicly available due to privacy.

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
