# Peer review of "A Systematic Analysis of the Correlation between Flavor Active Differential Metabolites and Multiple Bean Ripening Stages of Coffea arabica L."

_molecules, 2023, doi:10.3390/molecules29010180_

Round 1

Reviewer 1 Report

Comments and Suggestions for Authors

Overall, the manuscript is very well written. It deals with interesting issues and contains valuable data that might be usable for further reference. The manuscript contains minor grammatical errors and hard-to-read sentences. Therefore, it is suggested to proofread the manuscript. Some of the figures, i.e., Figure 2B is hard to read, and the resolution is, in my opinion, low.

Abstract

Line 17 and 18 flavor/flavor, please use the same style throughout the manuscript.

Line 20 please substitute „there were“ with other appropriate collocation

Introduction

Line 41: Please rephrase the first sentence.

Line 42: Only in developing countries? Please rephrase

Line 50: Please rephrase the sentence. It is not grammatically correct.

Line 57 – 58 The diversity of coffee quality is affected by the biochemical components of the green coffee bean accumulated during cherry development.

Line 59 „Chemically altered“  Please specify

Line 75 I would instead use „composition„ than „content“

Results:

Line 107: Please explain GF, ROF, BRF, PRF, and PBF in brackets. Since this is the beginning of the sections from my point of view it is necessary to explain them clearly, so the rest of the Results section will be easier to read.

Line 107: Please explain the „pre-treatment “. If necessary, state it in the manuscript or add a note that will help the reader find the section/subsection that deals with the explanation of the treatment.

Line 133 Altogether, how many factors were calculated in PCA analysis?

What test was used to determine the significance of groups separation? Please explain.

Why did the authors use PCA instead of LDA (linear discriminant analysis or NBC (Naive Bayes Classifier)? Please explain.

Line 248: error: „metabolitesuch“

Line 253 -254 Please rephrase the sentence. It is hard to read.

Line 275 What saccharides, please specify.

Line 278: Did the authors perform sensory analysis and compare it with data, or in this case, the literature was used? Please specify

Methodology: 

The authors did not state that the subsection focused on performed statistical analysis. I suggest adding it to the manuscript.

Comments on the Quality of English Language

Author Response

Dear reviewer 1,

Thanks for your useful suggestions on this paper. In response to your questions, we will address each point in turn in the following pages.

Abstract

  1. Line 17 and 18 flavor/flavor, please use the same style throughout the manuscript.

Answer: Thank you for your carefulness, we had some oversights in the word flavor, which have been corrected.

  1. Line 20 please substitute „there were“ with other appropriate collocation

Answer: We've changed “there were” to “we identified”, thanks for the suggestion!

Introduction

  1. Line 41: Please rephrase the first sentence.

Answer: We have rephrased the sentence.

  1. Line 42: Only in developing countries? Please rephrase

Answer: We have made a correction.

  1. Line 50: Please rephrase the sentence. It is not grammatically correct.

Answer: We have rephrased the sentence.

  1. Line 57 – 58 The diversity of coffee quality is affected by the biochemical components of the green coffee bean accumulated during cherry development.

Answer: We've replaced the sentence as you intended.

  1. Line 59 „Chemically altered“  Please specify

Answer: We have clarified its meaning by describing

  1. Line 75 I would instead use „composition„ than „content“

Answer: We have made a correction.

Results:

  1. Line 107: Please explain GF, ROF, BRF, PRF, and PBF in brackets. Since this is the beginning of the sections from my point of view it is necessary to explain them clearly, so the rest of the Results section will be easier to read.

Answer: We have made a detailed explanation in the sentence according to your suggestion, thanks.

  1. Line 107: Please explain the „pre-treatment “. If necessary, state it in the manuscript or add a note that will help the reader find the section/subsection that deals with the explanation of the treatment.

Answer: We have meticulously described the pre-treatment in the Materials and Methods 3.1, and this treatment is the standard water-washing method for coffee assays.

  1. Line 133 Altogether, how many factors were calculated in PCA analysis?

What test was used to determine the significance of groups separation? Please explain. Why did the authors use PCA instead of LDA (linear discriminant analysis or NBC (Naive Bayes Classifier)? Please explain.

Answer: A total of 18 principal components were calculated for the PCA analysis. Principal component 1 (PC1) reflects the differences between groups, and principal component 2 (PC2) responds to the differences in the samples between groups, as can be seen in the reference.

Ding, J., Ji, J., Rabow, Z., Shen, T., Folz, J., Brydges, C. R., Fan, S. L., Lu, X. C., Mehta, S., Showalter, M. R., Zhang, Y., Araiza, R., Bower, L. R., Lloyd, K. C. K., and Fiehn, O. A metabolome atlas of the aging mouse brain. Nature Communications 2021, 12, 12.

LDA and PCA: the biggest difference is that PCA is unsupervised clustering, LDA is supervised clustering

(1) LDA requires that in the process of dimensionality reduction can be used in the category of prior knowledge experience, while PCA does not need to have prior knowledge

(2) LDA dimensionality reduction up to the number of categories k-1 dimensions, if we reduce the dimensionality is greater than k-1, then we can not use LDA

(3) LDA may overfitting

NBC: The core idea is to give features and then obtain categories. Moreover, this method requires that features be independent, while metabolic substances contain upstream and downstream relationships, which is not consistent. In this paper, we only use PCA to evaluate the quality of data, not to do classification, so we don't use NBC. In addition, PCA is the most commonly used to evaluate metabolic articles in the sample group and between groups can separate the most common dimension reduction method.

  1. Line 248: error: „metabolitesuch“

Answer: We have made a correction.

  1. Line 253 -254 Please rephrase the sentence. It is hard to read.

Answer: We have made a correction.

  1. Line 275 What saccharides, please specify.

Answer: We have noted that, thank you very much.

  1. Line 278: Did the authors perform sensory analysis and compare it with data, or in this case, the literature was used? Please specify

Answer: This was an error in the presentation of the paper and we have corrected it accordingly. The paper selected typical coffee taste indicators for correlation analysis, and we did not cup test the coffee samples because of the insufficient number of samples.

Methodology: 

  1. The authors did not state that the subsection focused on performed statistical analysis. I suggest adding it to the manuscript.

Answer: We have added statistical analysis section to the methodology as you intended, thank you very much for your suggestion!

In addition, we have modified Figure 2B at 1200ppi. All in all, thank you very much for your valuable comments.

Kind regards!

Reviewer 2 Report

Comments and Suggestions for Authors

The manuscript “Systematic Analysis of Correlation between Flavor Active Differential Metabolites and Multiple Bean Ripening Stages of Coffea arabica L” presents a systematic analysis of the correlation between flavor active differential metabolites and multiple bean ripening stages of Coffea arabica L. The study analyzes the flavor and active substances in green coffee beans during cherry ripening, providing valuable reference information to clarify the formation of substances in Coffea arabica L. The research sheds light on the importance of selecting the appropriate cherry maturity level in the coffee manufacturing process. The major impact of this study is that it provides valuable information for coffee manufacturers to improve the coffee manufacturing process and ensure consistent flavor quality.

Some important corrections:

·       Only around 21% of the references are from the last 5 years – the authors should update the reference section as it highlights its actuality

·       Please shorten and make the abstract section more concise, be attentive to the author's guidelines as this section should have a maximum of 200 words

· A native English speaker should revise the manuscript

·       Line 95 – assuming that there is no discussion section it should be corrected to Results and Discussions

·       Line 312 – the sentence “The C. arabica cultivar catimor used as test material in this study.” Is not correct, please revise

·       Line 328 – please specify the country of production for each instrument used (i.e. freeze-dryer, grinder, etc)

·       Please indicate the limitations of the study

·       The conclusion should be moved to the end of the results and discussion section as it summarizes the main findings of the study

·       For the conclusion section the authors should provide a clear and concise conclusion that ties together the main points of the study and emphasizes its importance. Also, the authors should suggest areas for future research

 Overall, this study provides valuable insights into the nutritional composition of coffee and the accumulation patterns of biochemical components in green coffee beans, which can be useful for further understanding the factors that contribute to the flavor and aroma of coffee and for coffee breeding research. After some major revisions, it could be considered for publication.

Comments on the Quality of English Language

A native English speaker should revise the manuscript

Author Response

Dear reviewer 2,

Thanks for your useful suggestions on this paper. In response to your questions, we will address each point in turn in the following pages.

  1. Only around 21% of the references are from the last 5 years – the authors should update the reference section as it highlights its actuality

Answer: We tried our best to update some of the newly published paper on the subject as you intended, but since some of the paper is more classic in terms of coffee maturity research, we were not able to remove them very smoothly. Overall, thank you very much for your valuable suggestions.

  1. Please shorten and make the abstract section more concise, be attentive to the author's guidelines as this section should have a maximum of 200 words.A native English speaker should revise the manuscript

Answer: We made the changes as best we could, but because some very necessary information could not be removed, we were only able to keep it to about 270 words.

  1. Line 95 – assuming that there is no discussion section it should be corrected to

Answer: We have made a correction.

Results and Discussions

Answer: We have made a correction.

  1. Line 312 – the sentence “The C. arabica cultivar catimor used as test material in this study.” Is not correct, please revise

Answer: We have made a correction.

  1. Line 328 – please specify the country of production for each instrument used (i.e. freeze-dryer, grinder, etc)

Answer: We've added descriptive information about the instrument. Thank you!

  1. Please indicate the limitations of the study

Answer: We have added limitations and outlooks at the end of the Results and Discussions, thanks for your suggestion.

  1. The conclusion should be moved to the end of the results and discussion section as it summarizes the main findings of the study

Answer: We have moved the conclusion in the paper.

  1. For the conclusion section the authors should provide a clear and concise conclusion that ties together the main points of the study and emphasizes its importance. Also, the authors should suggest areas for future research

Answer: We have added the outlooks at the end of the Results and and Discussions. This passage-However, the impact of ripening-induced changes in these compounds on baked beans remains unknown and requires further investigation. Additionally, it is essential to enhance our understanding of diversity by considering different coffee varieties with distinct flavor characteristics and varying levels of flavor precursors in green coffee beans.

All in all, thank you very much for your valuable comments.

Kind regards!

Round 2

Reviewer 2 Report

Comments and Suggestions for Authors

The manuscript has been considerably improved. 

I still consider that the abstract section could be shortened to make it more concise such as:

The coffee cherry contains crucial flavor-precursor and chemical substances influencing roasted bean quality, yet limited knowledge exists on metabolite changes during cherry ripening. Our study identified 1078 metabolites, revealing 46 core differential metabolites through KEGG pathway analysis. In the GF vs. ROF stage, amino acid synthesis dominated; ROF vs. BRF featured nucleotide catabolism; BRF vs. PRF exhibited glycoside and flavonoid synthesis, and PRF vs. PBF involved secondary metabolite synthesis and catabolism. The PRF stage emerged as the optimal cherry harvesting period. Correlation analysis identified core differential metabolites strongly linked to taste indicators, suggesting their potential as taste markers. Notably, nucleotides and derivatives exhibited significant negative correlations with glycosides and flavonoids during ripening. This research systematically analyzed flavor and active substances in green coffee beans during cherry ripening, offering valuable insights into substance formation in Coffea arabica L.

 Otherwise the manuscript has been considerably improved and it can be considered for publication

Author Response

Dear reviewer 2,

As your said,

I still consider that the abstract section could be shortened to make it more concise such as:

The coffee cherry contains crucial flavor-precursor and chemical substances influencing roasted bean quality, yet limited knowledge exists on metabolite changes during cherry ripening. Our study identified 1078 metabolites, revealing 46 core differential metabolites through KEGG pathway analysis. In the GF vs. ROF stage, amino acid synthesis dominated; ROF vs. BRF featured nucleotide catabolism; BRF vs. PRF exhibited glycoside and flavonoid synthesis, and PRF vs. PBF involved secondary metabolite synthesis and catabolism. The PRF stage emerged as the optimal cherry harvesting period. Correlation analysis identified core differential metabolites strongly linked to taste indicators, suggesting their potential as taste markers. Notably, nucleotides and derivatives exhibited significant negative correlations with glycosides and flavonoids during ripening. This research systematically analyzed flavor and active substances in green coffee beans during cherry ripening, offering valuable insights into substance formation in Coffea arabica L.

Answer: Thank you very much for your pertinent and useful suggestions, and we have replaced the summary section as you intended.

All in all, thanks you again for your constructive comments and hard work on this article, and even for helping me write a number of sentences that we'll replace as you intended. Best wishes to you!

Kind regards!